

# Integrative genomic and transcriptomic analyses of a bud sport mutant 'Jinzao Wuhe' with the phenotype of large berries in grapevines

Jianquan Huang[1,*], Guan Zhang[2,3,*], Yanhao Li[1,4], Mingjie Lyu[2], He Zhang[1], Na Zhang[1] and Rui Chen[2]

[1] The Research Institute of Forestry and Pomology, Tianjin Academy of Agricultural Sciences, Tianjin, China
[2] Institute of Crop Germplasm and Biotechnology, Tianjin Academy of Agricultural Sciences, Tianjin, China
[3] College of Biotechnology and Food Science, Tianjin University of Commerce, Tianjin, China
[4] College of Horticulture and Gardening, Tianjin Agricultural University, Tianjin, China
* These authors contributed equally to this work.

Corresponding authors
Na Zhang, 2371254838@qq.com
Rui Chen, chenrui_aglab@126.com

## ABSTRACT

**Background:** Bud sport mutation occurs frequently in fruit plants and acts as an important approach for grapevine improvement and breeding. 'Jinzao Wuhe' is a bud sport of the elite cultivar 'Himord Seedless' with obviously enlarged organs and berries. To date, the molecular mechanisms underlying berry enlargement caused by bud sport in grapevines remain unclear.

**Methods:** Whole genome resequencing (WGRS) was performed for two pairs of bud sports and their maternal plants with similar phenotype to identify SNPs, InDels and structural variations (SVs) as well as related genes. Furthermore, transcriptomic sequencing at different developmental stages and weighted gene co-expression network analysis (WGCNA) for 'Jinzao Wuhe' and its maternal plant 'Himord Seedless' were carried out to identify the differentially expressed genes (DEGs), which were subsequently analyzed for Gene Ontology (GO) and function annotation.

**Results:** In two pairs of enlarged berry bud sports, a total of 1,334 SNPs, 272 InDels and 74 SVs, corresponding to 1,022 target genes related to symbiotic microorganisms, cell death and other processes were identified. Meanwhile, 1,149 DEGs associated with cell wall modification, stress-response and cell killing might be responsible for the phenotypic variation were also determined. As a result, 42 DEGs between 'Himord Seedless' and 'Jinzao Wuhe' harboring genetic variations were further investigated, including pectin esterase, cellulase A, cytochromes P450 (CYP), UDP-glycosyltransferase (UGT), zinc finger protein, auxin response factor (ARF), NAC transcription factor (TF), protein kinase, *etc.* These candidate genes offer important clues for a better understanding of developmental regulations of berry enlargement in grapevine.

**Conclusion:** Our results provide candidate genes and valuable information for dissecting the underlying mechanisms of berry development and contribute to future improvement of grapevine cultivars.

# INTRODUCTION

Grape (*Vitis vinifera* L.) is one of the most favorite fruits worldwide with great economic and nutritional value (*Yang & Xiao, 2013*). Different from routine improvement strategies in crops, grapevine is highly heterozygous and exhibit serious genetic segregation, which hinders the development process of excellent traits. So far, various breeding techniques had been applied to improve the genetic background of grapes and achieve reliable and stable inheritance of desirable traits (*Yamada & Sato, 2016*). Out of these approaches, bud sport is an important strategy for grapevine improvement and breeding. Bud sport mutation is a phenomenon usually happened in fruit plants in which stable somatic cells are mutated in the meristematic tissue of the bud and exhibit novel traits due to endogenous and exogenous factors. In this process, favorable phenotypes caused by mutation (*e.g.*, shorter ripening cycle, larger berries, and different skin color) will be selected by breeders that are significantly different from the maternal plants with desired and superior characteristics (*Foster & Aranzana, 2018*). To date, numerous bud sport mutants have been observed and successfully utilized in grapevine breeding, such as early-ripening cultivars 'Fengzao', 'Tiangong Moyu', and 'Nantaihutezao' (*Guo & Zhang, 2015*; *Wei et al., 2020*; *Leng et al., 2021*), as well as berry skin color mutants 'Pinot Grigio', 'Pinot Blanc' and 'Benitaka' (*Azuma et al., 2009*; *Pan et al., 2012*; *Negri et al., 2015*). However, the phenotype with large berries induced by bud sport has not been reported and the corresponding regulatory mechanisms remain poorly understood.

Berry size is a polygenic quantitative trait in grapevine. Similar to other fruit plants, grapevine follows a double sigmoid curve corresponding to the fruit enlargement, véraison and maturity stages. Few days after flowering, berry development occurs through cell division and expansion. During the enlargement stage, the berry grows slowly and ends with the véraison stage, where it softens and the color of the skin changes. In the véraison stage, berry growth restarts but exclusively through cell enlargement. In the maturity stage, all the tissues that make up the berries are formed. Previous studies in Arabidopsis (*Krizek, 1999*; *Wang et al., 2016*), apple (*Malladi & Hirst, 2010*), tomato (*Su et al., 2014*) and peach (*Scorzal et al., 1991*) indicated that cell division and cell expansion play crucial roles in controlling fruit size.

Several key genes and regulators had been demonstrated to be responsible for berry enlargement which involved in cell wall modification, phytohormone production, water and sugar transport processes. "Pectin Lyase (PL)", "wall–associated receptor kinase (WAK)" and "Pectin Esterase (PE)" are related with the accumulation of pectin, that can directly bind pectin polymers to reduce stiffness and stimulate cell wall development (*Tucker et al., 2018*). Ethylene was previously thought to have limited effects on grape growth and development, that is why grapevines were classified as non-climacteric berries (*Chervin et al., 2004*). Recent evidence suggested that ethylene coupled with increased ACO transcription could produce a modest peak during berry ripening and promote larger berry diameter, higher sugar accumulation (*De La Torre-Ruiz et al., 2016*), and higher

anthocyanin synthesis (*El-kereamy et al., 2003*). UGT, as sugar donors, are primarily responsible for the formation of secondary metabolites such as antibiotics, flavanols, and steroids through glycosylation modifications that affect plant growth and development (*Kiedrowski et al., 1992*; *Mitchell, Hall & Barber, 1994*; *Breton, Fournel-Gigleux & Palcic, 2012*). Cytochrome P450 (CYP) is a highly conserved enzyme family that plays significant roles in plant development and growth by increasing cell size, responding to biotic and abiotic stresses, and stimulating phytohormone signal transduction (*Ma et al., 2015*). *PaCYP78A9* was inferred to be involved in the regulation of organ size in sweet cherry (*Qi et al., 2017*). CYPs, such as *EOD3/CYP78A6* and *CYP78A9* in Arabidopsis (*Fang et al., 2012*;), *CYP78A5* in tomato (*Chakrabarti et al., 2013*), *GmCYP78A10* in soybean (*Wang et al., 2015*), *CYP78A13* (*Xu et al., 2015*) and *GW10* in rice (*Zhan et al., 2021*), have all been identified as seed size regulators.

Transcription factors (TFs) are also important regulators in controlling berry size which are typically transcribed in response to hormonal and environmental stimuli. NAC transcription factor has been proven to be functional in fruit development and maturation (*Migicovsky et al., 2021*). For example, the participation of *VvNAC26* determinates the berry ultimate size in grapevine (*Tello et al., 2015*). Overexpression of *VvCEB1* and *VvSAUR41* genes in grapevine embryos increase fruit diameter by stimulating cell expansion (*Nicolas et al., 2013*; *Li et al., 2021*). *VvANT1*, an ortholog of Arabidopsis ANT, could regulate cell proliferation and organ expansion (*Chialva et al., 2016*). In addition, *Martins et al., (2018)* have demonstrated that calcium-related genes have a significant regulatory effect on the cell wall biogenesis by influencing the expression of pectin methyl esterase (PME) and polygalacturonase (PG).

In this study, 'Jinzao Wuhe' is a very rare bud mutation discovered and developed by our laboratory due to its remarkably enlarged berry size. Herein, we carried out WGRS and genetic variation analysis for 'Jinzao Wuhe' and its maternal cultivar to explore the candidate genes in response to berry development, as well as the potential mechanisms underlying bud sport in grapevines. Furthermore, transcriptome sequencing and WGCNA analyses were performed at four developmental time-points, leading to the identification of causal genes and regulators. These results would shed light on the molecular mechanisms of bud sport mutation and offer useful information for future grapevine improvement.

## MATERIALS AND METHODS

### Plant materials

'Himord Seedless' is an early maturing cultivar with seedless and small berries, ripening before the onset of the rainy season in Tianjin, China. 'Jinzao Wuhe' is a bud sport derived from 'Himord Seedless' discovered at Tianjin Academy of Agricultural Sciences (Tianjin, China) orchard in 2014. The 3-year-old vines 'Himord Seedless' and 'Thompson Seedless' and their bud sports 'Jinzao Wuhe' and 'Dawuhebai' were planted in the grape garden of Tianjin Academy of Agricultural Sciences, Tianjin, China, grown on Y-shaped trellises with 1.5 m (column) × 3 m (row) interval. Management guidelines include techniques for irrigation, fertilization, pruning, weeding, disease prevention, pest control and precautions during rainy season. Compared with its maternal plant 'Himord Seedless', 'Jinzao Wuhe'

exhibits larger leaves, inflorescences, berries, thicker stems, roots, and greater vigorous growth compared to its maternal plant. Especially, the berry size of 'Jinzao Wuhe' is almost twice as large as its maternal plant, which is an excellent germplasm resource for grapevine breeding. In addition, 'Dawuhebai', another bud sport derived from table grape 'Thompson Seedless', also possesses enlarged berries that would be helpful for investigating the regulatory mechanisms underlying berry enlargement (*Baiano & Terracone, 2012*).

The longitudinal diameter, transverse diameter, volume and weight of each ten berries for each replicate were measured individually for every 7 days started from the 10 days post-anthesis (DPA). Three biological replicates were set for each treatment. For each grapevine cultivar, 2-week-old leaves without pathogens and insect infestations were harvested for genomics DNA isolation, library construction and WGRS. Inflorescences and berries of 'Himord Seedless' and 'Jinzao Wuhe' that were healthy and vigorous were sampled at the full-flowering stage (stage 1: 1 DPA), berry expansion stage (stage 2: 14 DPA), véraison stage (stage 3: 35 DPA) and maturity stage (stage 4: 56 DPA) for RNA extraction and transcriptome sequencing, respectively. Three biological replicates were prepared for each sampling time point. These leaf, inflorescence and berry samples were immediately snap-frozen by immersion in liquid nitrogen and stored at −80 °C until use.

## DNA and RNA extraction and sequencing

Genomic DNA was extracted from grapevine leaves by using the modified protocol of cetyl trimethyl ammonium bromide (CTAB). The concentration and purity of the extracted genomic DNA was measured by NanoDrop-1000 spectrophotometer (NanoDrop Technologies, Wilmington, DE, USA) and the quality and integrity of isolated genomic DNA was assessed by 1% agarose gel electrophoresis. Paired-end (PE) Deep sequencing libraries were prepared according to the standard protocol (Illumina) with the Paired-end (PE) strategy. Finally, the qualified libraries were sequenced using the Illumina HiSeq 2500 platform in Novogene Biotechnology Company (Tianjin, China). Raw data was deposited in the NCBI SRA database under the BioProject accession PRJNA795039 and PRJNA872374 (Table S1).

Total RNA was isolated from inflorescences and berries of 'Himord Seedless' and 'Jinzao Wuhe' at 1, 14, 35, and 56 DPA using TRIzol reagent according to the manufacturer's instructions. The quality of RNA was evaluated by NanoDrop-1000 spectrophotometer (NanoDrop Technologies, Wilmington, DE, USA), and RNA integrity was determined by 1% denaturing gel electrophoresis. Three replicates at different stages were set for each sample. RNA-seq libraries were then prepared with the Illumina TruSeq RNA Sample Prep Kit and sequenced by using the Illumina HiSeq X Ten System in Novogene Biotechnology Company (Tianjin, China). All raw RNA-seq data were deposited in the NCBI SRA database under BioProject accession PRJNA795246 (Table S1).

## SNP, InDel and SV calling and annotation

Illumina raw reads were filtered and trimmed for low quality bases and adapter sequences using Fastp (default parameters), and then mapped onto the reference genome of *V. vinifera* Pinot Noir PN40024 (12X, GCF_000003745.3) using BWA (version 0.7.17) (*Li*

& Durbin, 2009) with default parameters. Marking duplicated sequences caused by PCR reactions using the Genome Analysis Toolkit (GATK4) (*McKenna et al., 2010*) with the MarkDuplicates module (default parameters). Both SNPs and InDels were identified for each sample using the GATK4 pipeline with the HaplotypeCaller module according to the recommended variation detection workflow (gatk –java-options HaplotypeCaller-R $genome_file-ERC GVCF-I $bam-O $out). The criteria for variant filtration were as follows: (i) biallelic SNPs and InDels were retained; (ii) sequencing depth of higher than 5 was selected; (iii) variant allele frequencies of less than 40% were excluded; (iv) missing genotypes were deleted; (v) InDels shorter than or equal to 50 bp were retained.

Structural variations (SVs) were detected in all samples using LUMPY (version 0.2.13) (*Layer et al., 2014*) and DELLY (version 0.8.7) (*Rausch et al., 2012*). Splitters and discordant reads were extracted by using Lumpyexpress (lumpyexpress-B $sample.bam-S $sample.splitters.bam-D $sample.discordants.bam-o lumpyout_single_$tag.vcf) for each individual sample. Bam files were sorted and indexed using Samtools (version 1.12) (*Li et al., 2009*) and SVs were genotyped using SVTyper software. For DELLY (version 0.8.7), SVs were presented with the recommended instructions in BCF format (delly call-t ALL-g $genome $sample.bam-o $sample.bcf), which were converted to VCF format using BCFtools (bcftools view $sample.bcf > $sample.vcf) (*Danecek & McCarthy, 2017*). Moreover, SURVIVOR (SURVIVOR merge) was used to merge SVs generated from both LUMPY and DELLY pipelines. As for filtration, SVs with a length of 50 bp or more and the quality flag PASS, as well as the precise breakpoints, were selected as the final dataset for further analysis.

The intersection of SNP, InDel, and SV datasets of the two pairs of bud mutation and their maternal were selected separately. Then, the effect of all selected variants located in chromosomes was annotated using Snpeff (java-jar snpEff.jar eff-csvStats $sample.csv-s $sample.html-c snpEff.config-v grape $sample.vcf.gz >$sample_snpeff.gz) (*Cingolani et al., 2012*), which classified them into HIGH, LOW, MODERATE, and MODIFIER categories according to the predicted results for influencing gene function. Important and non-synonymous SNPs, InDels and SVs with high impact were selected as target variants that might be responsible for phenotype variation.

## Transcriptome analysis

Tophat-Cufflinks pipeline was used to generate gene expression profiles and identify DEGs (*Trapnell, Pachter & Salzberg, 2009*). First of all, raw reads were preprocessed using Fastp (default parameters) (*Chen et al., 2018*) to remove adaptors and low quality bases. Tophat2 (default parameters) was carried out for genomic mapping. Gene expression levels were estimated as fragments per kilobase of transcript per million (FPKM) by Cufflink (default parameters) based on the alignments. DEGs between different developmental stages were identified using Cuffdiff with regarding a false discovery rate (FDR) ≤0.05 and log2 Fold-Change ≥ 1 as thresholds. Meanwhile, HiSAT2 (*Kim, Langmead & Salzberg, 2015*), feature counts (*Liao, Smyth & Shi, 2014*) and DESeq2 (*Love, Huber & Anders, 2014*, p. 2) workflows were also performed to achieve a more comprehensive result of DEGs analysis. The statistical power calculated using RNASeqPower (Bioconductor—RNASeqPower) was
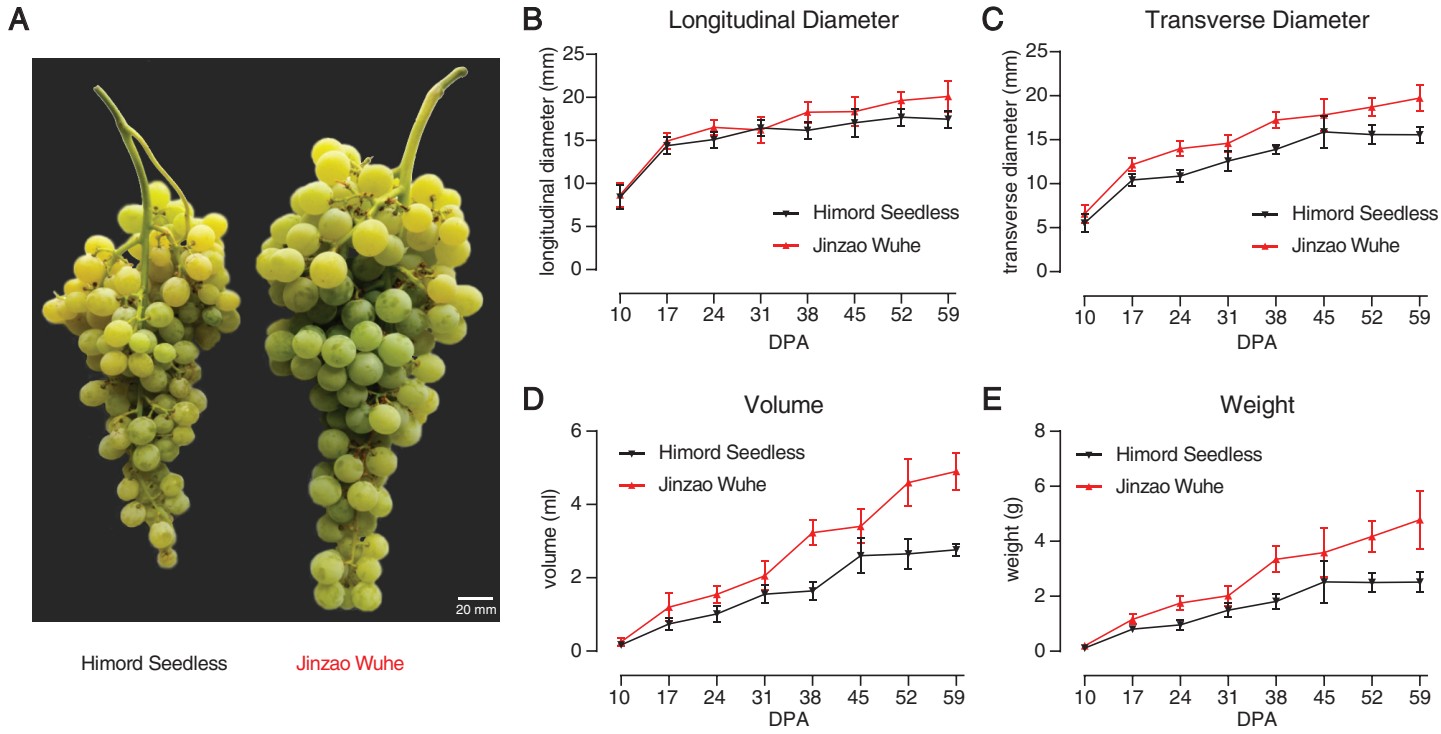

**Figure 1** **The comparison of 'Himord Seedless' and its bud sport 'Jinzao Wuhe'.** Comparison of longitudinal diameter (B), transverse diameter (C), volume (D) and weight (E) of 'Himord Seedless' (black line) and 'Jinzao Wuhe' (red line) at different developmental stages. Standard deviations (SDs) were calculated with 30 biological replicates for each timepoint. The error bars represented 1 SD.

0.96. We visualized the correlation among the three biological replicates for each sample and gene expression level using the R package 'scatterplot3d' and 'pheatmap', respectively.

## WGCNA network analysis

WGCNA was carried out in R program with the WGCNA package (*Langfelder & Horvath, 2008*). Following the WGCNA tutorial, the FPKM values of DEGs were chosen for further investigation. The soft-thresholding power (β) was determined using the "pickSoftThreshold" function. Hierarchical cluster tree displaying co-expression modules was constructed as follows: minimodule size = 50, deep split = 2; merge cut height = 0.35. To identify the relationship between co-expression modules and phenotypes, the module-trait relationships were subsequently determined by Pearson correlation.

## RESULTS

### Phenotypical comparison between 'Himord Seedless' and its bud sport mutant 'Jinzao Wuhe'

Physiological traits were measured, where 'Jinzao Wuhe' showed a double sigmoid pattern curve from fruiting (10 DPA) to maturity (59 DPA) with higher berry longitudinal diameter, transverse diameter, volume and weight than 'Himord Seedless' (Figs. 1B–1E; Table S2). Two discriminable stages for berry expansion could be observed during berry

**Table 1  The statistics of whole-genome resequencing data.**

| Sample | Raw reads | Clean reads | Q20 | Q30 | Mapping rates |
|---|---|---|---|---|---|
| Thompson seedless | 54,003,664 | 53,434,316 | 95.52% | 88.93% | 93.57% |
| Dawuhebai | 75,630,160 | 75,128,802 | 93.57% | 90.50% | 95.87% |
| Himord seedless | 91,982,400 | 91,164,032 | 96.30% | 90.52% | 91.91% |
| Jinzao Wuhe | 83,784,034 | 83,013,230 | 93.70% | 90.19% | 95.39% |

growth. At the end of maturity (59 DPA), 'Jinzao Wuhe' was about twice as large as the 'Himord Seedless' in terms of volume and weight (Fig. 1A).

## SNP, InDel and SV calling

To identify the genome-wide genetic variations between two maternal grapevines (WTs: 'Thompson Seedless' and 'Himord Seedless') and their bud mutants (MUTs: 'Dawuhebai' and 'Jinzao Wuhe'), WGRS was performed, producing a total of 305,400,258 raw reads (~45.8 Gb). After removing the low-quality reads and bases, a total of 302,740,380 clean reads (~45.5 Gb) were retrieved, accounting for more than 99.34% of the raw data. The Q20 and Q30 Phred Scores of all samples were greater than 93.57% and 88.93%, respectively (Table 1). The genomic mapping rates for clean reads ranged from 91.91% to 95.87% (Table 1).

For 'Thompson Seedless', 'Dawuhebai', 'Himord Seedless' and 'Jinzao Wuhe', 3,763,122, 3,947,775, 5,512,478 and 5,499,289 SNPs were detected, respectively. As a result, MUTs are richer in SNPs than WTs and the number of SNPs of 'Jinzao Wuhe' is higher than 'Dawuhebai' (Fig. 2A). We also discovered 2,658,536, 3,663,339, 2,531,663 and 3,670,669 transitions (Ts) and 1,289,239, 1,835,950, 1,231,459 and 1,841,809 transversions (Tv) in 'Thompson Seedless', 'Dawuhebai', 'Himord Seedless', and 'Jinzao Wuhe', respectively, with genome-wide transitions to transversions ratio (Ts/Tv) were 2.06, 2.00, 2.06, and 1.99 (Table S3). SnpEff annotation revealed that the SNPs were mainly located in the upstream and downstream of genes, followed by intergenic regions, intronic regions, and exon regions, while they were scarce in splice site regions, UTR3' and UTR5' regions. In total, the heterozygous rates in 'Himord Seedless' and 'Jinzao Wuhe' were apparently higher than those in 'Thompson Seedless' and 'Dawuhebai' (Table S3).

'Thompson Seedless', 'Dawuhebai', 'Himord Seedless' and 'Jinzao Wuhe' have 698,381 (insertion: 382,040; deletion: 316,341), 752,785 (insertion: 341,594; deletion: 411,191), 1,113,041 (insertion: 516,558; deletion: 596,483) and 1,108,155 (insertion: 513,699; deletion: 594,456) InDels, respectively. It could be inferred that 'Himord Seedless' and 'Jinzao Wuhe' have more distant genetic relationships with the reference genome (Table S4). Besides, the length distribution of InDels showed that mononucleotide insertions or deletions accounted for the majority of InDels (Fig. 2B). The frequency of InDels located in the upstream and downstream of genes was higher, averaging at 29.30% and 30.26%. Approximately 15.00%, 11.00%, and 1.00% of InDels harbored in intergenic, intronic, and exon regions, respectively. And they were almost undetectable for the splice
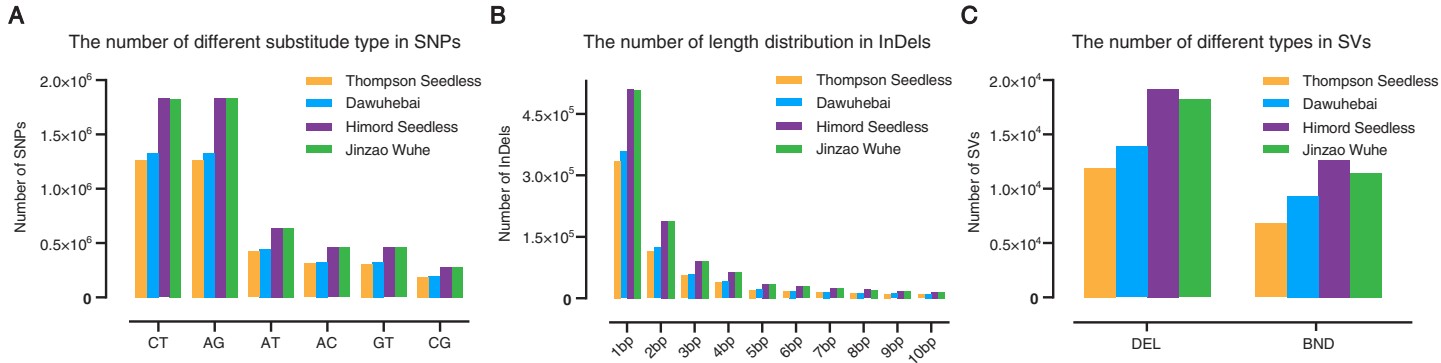

**Figure 2** Annotation and distribution of (A) SNPs, (B) InDels and (C) SVs of 'Thompson Seedless' and its bud sports 'Dawuhebai', 'Himord Seedless' and its bud sports 'Jinzao Wuhe'.

site, UTR3'and UTR5' regions (all less than 1.00%). In addition, homozygous and heterozygous InDels in four grapevine samples were also calculated, with the heterozygous rate of 79.32% and 79.19% in 'Himord Seedless' and 'Jinzao Wuhe', respectively, which were higher than 68.02% and 70.50% in 'Thompson Seedless' and 'Dawuhebai', respectively (Table S4).

SVs play an important functional role in genetic and morphological divergence, mainly including deletions (DEL), translocation (BND), inversions (INV), duplications (DUP) and insertions (INS). Due to the limitations of Illumina sequencing platform that usually producing short reads less than 200 bp, the results of DUP, INV, and INS were excluded for further investigation. Only DEL and BND were remained for subsequent analysis. In 'Thompson Seedless', 'Dawuhebai', 'Himord Seedless' and 'Jinzao Wuhe', 18,717, 23,237, 31,807 and 29,715 SVs were identified, respectively. The number of SVs in MUTs was markedly higher than in WTs. The distribution of SVs of four grape lines showed that the proportion of deletions was more than 53.73%, whereas other SVs accounted for less than 46.27%. (Fig. 2C). In summary, the distribution and proportions of SVs among these four grapevine lines were highly similar to those of SNPs and InDels (Table S5).

## Identification of candidate genes harboring SNPs, InDels and SVs

To identify potential causal genes that are affected by DNA variations, comparative analysis between WTs and their MUTs were performed firstly. For 'Thompson Seedless' and 'Dawuhebai', 867,546, 184,194 and 21,817 different SNPs, InDels and SVs, as well as 3,535,491, 633,486 and 10,066 common SNPs, InDels and SVs were identified. For 'Himord Seedless' and 'Jinzao Wuhe', 706,821, 216,022 and 30,521 different SNPs, InDels and SVs, as well as 5,152,473, 1,002,587 and 15,497 common SNPs, InDels and SVs were identified, respectively (Figs. 3A–3C). Then, we compared the genetic variants between WTs and MUTs individually. Between 'Thompson Seedless' and 'Himord Seedless', 2,682,997 SNPs, 456,929 InDels and 8,250 SVs were regarded as common genetic variants. Between 'Dawuhebai' and 'Jinzao Wuhe', 2,775,884 common SNPs and 481,550 common InDels and 9,000 common SVs were identified (Figs. 3A–3C).

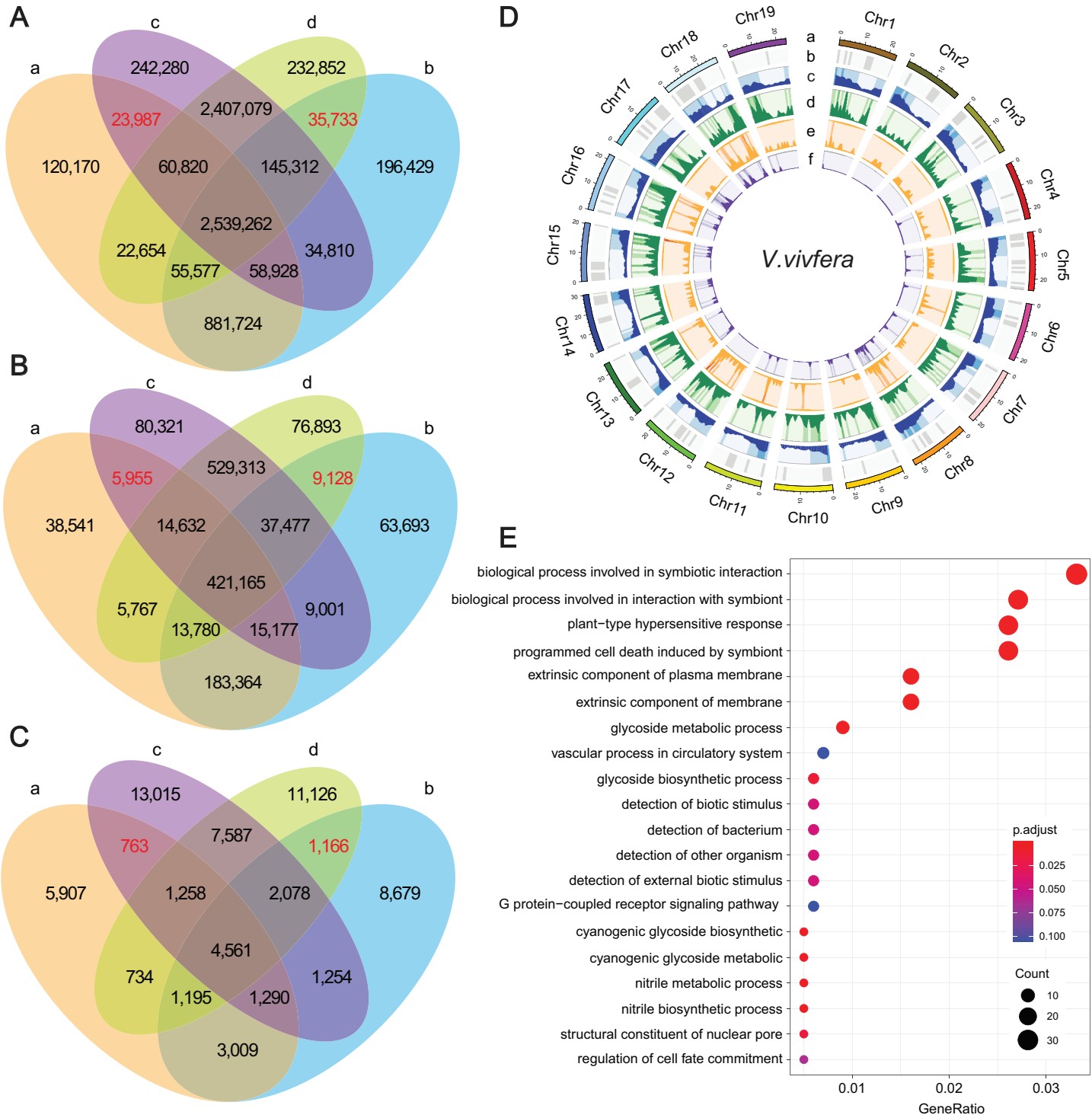

**Figure 3** **The statistics of SNPs, InDels and SVs and corresponding genes.** (A–C) Venn diagrams represent the numbers of unique and shared SNPs, InDels and SVs, respectively. (a) 'Thompson Seedless'; (b) 'Dawuhebai'; (c) 'Himord Seedless'; (d) 'Jinzao Wuhe'. (D) The circos plots of SNPs, InDels and SVs distribution on the nineteen *V.vivfera* chromosomes (a) Chromosome; (b) GC content; (c) Gene density; (d) SNP density; (e) InDel density; (f) SV density. (E) The GO annotation of potential causal genes harbored with determining SNPs, InDels and SVs.

These common variants derived from two independent bud sport mutation events could offer crucial clues for explaining the phenotypic variation in MUT berries that growing larger. Finally, 35,733 SNPs, 9,128 InDels and 1,166 SVs that were absent in WTs but co-existed in MUTs, as well as 23,987 SNPs, 5,955 InDels and 763 SVs that were absent in MUTs but co-existed in WTs were deemed as target variants that caused by bud sport mutation (Figs. 3A–3C). According to the results of snpEff analysis, 61 SNPs, 272 InDels and 74 SVs were predicted to have HIGH impact, whereas 1,273 SNP entries (missense variants) to have MODERATE impact (Table S6). These variants were distributed unevenly across 19 chromosomes, with SNPs and InDels predominating on Chr18 and Chr12, and SVs predominating on Chr 18 and Chr5 (Fig. 3D).

As a result, a total of 1,022 unique genes were finally identified based on these determining genetic variations, in which 891, 171 and 68 genes were associated with SNPs, InDels and SVs, respectively (Table S6). GO enrichment analysis of these genes showed 20 representative GO terms, including "biological process involved in symbiotic interaction", "biological process involved in interaction with symbiont", "plant-type hypersensitive response", "programmed cell wall", "glycoside metabolic process", "detection of biotic stimulus", "detection of external biotic stimulus", "structural constituent of nuclear pore", "regulation of cell fate commitment" *etc.* (Fig. 3E).

## Identification of the DEGs between 'Himord Seedless' and 'Jinzao Wuhe'

In order to obtain evidence at the transcriptomic level, RNA-seq was carried out and DEGs were investigated at four stages during berry development for 'Himord Seedless' and its mutant 'Jinzao Wuhe'. There was a good correlation among the three biological replicates for each sample (Fig. 4A). According to the analysis results of Cuffdiff, 928 (309 up-regulated and 619 down-regulated), 841 (358 up-regulated and 483 down-regulated), 1,627 (846 up-regulated and 781 down-regulated) and 727 (548 up-regulated and 179 down-regulated) DEGs were identified at stages 1–4, respectively (Figs. 4B, 4C, 4E and 4F; Table S7). After the redundancy removal, a total of 3,321 DEGs were identified that differentially expressed at least one time point (Fig. 4D). To achieve a more reliable result, another pipeline for RNA-seq and DEG analysis was performed using the same datasets. The results of DESeq2 generated 995 (346 up-regulated and 649 down-regulated), 1,080 (425 up-regulated and 655 down-regulated), 2,055 (1,125 up-regulated and 930 down-regulated) and 999 (733 up-regulated and 266 down-regulated) DEGs at stages 1–4, respectively (Table S8). As a result, 2,984 DEGs were finally confirmed by coupling two separate bioinformatic pipelines.

The GO enrichment analysis of theses DEGs identified 20 representative GO terms at four berry development stages. The terms were related to cell wall modification, such as "plant-type cell wall modification", "cell wall modification", "callose deposition in cell wall", "cell wall thickening", "pectin esterase activity" in all stages, which could provide a reasonable explanation between cell wall biogenesis and berry enlargement. Several GO terms were associated with stress-response, such as "response to heat", "response to toxic substance", "response to antibiotic", "response to reactive oxygen", "response to auxin",

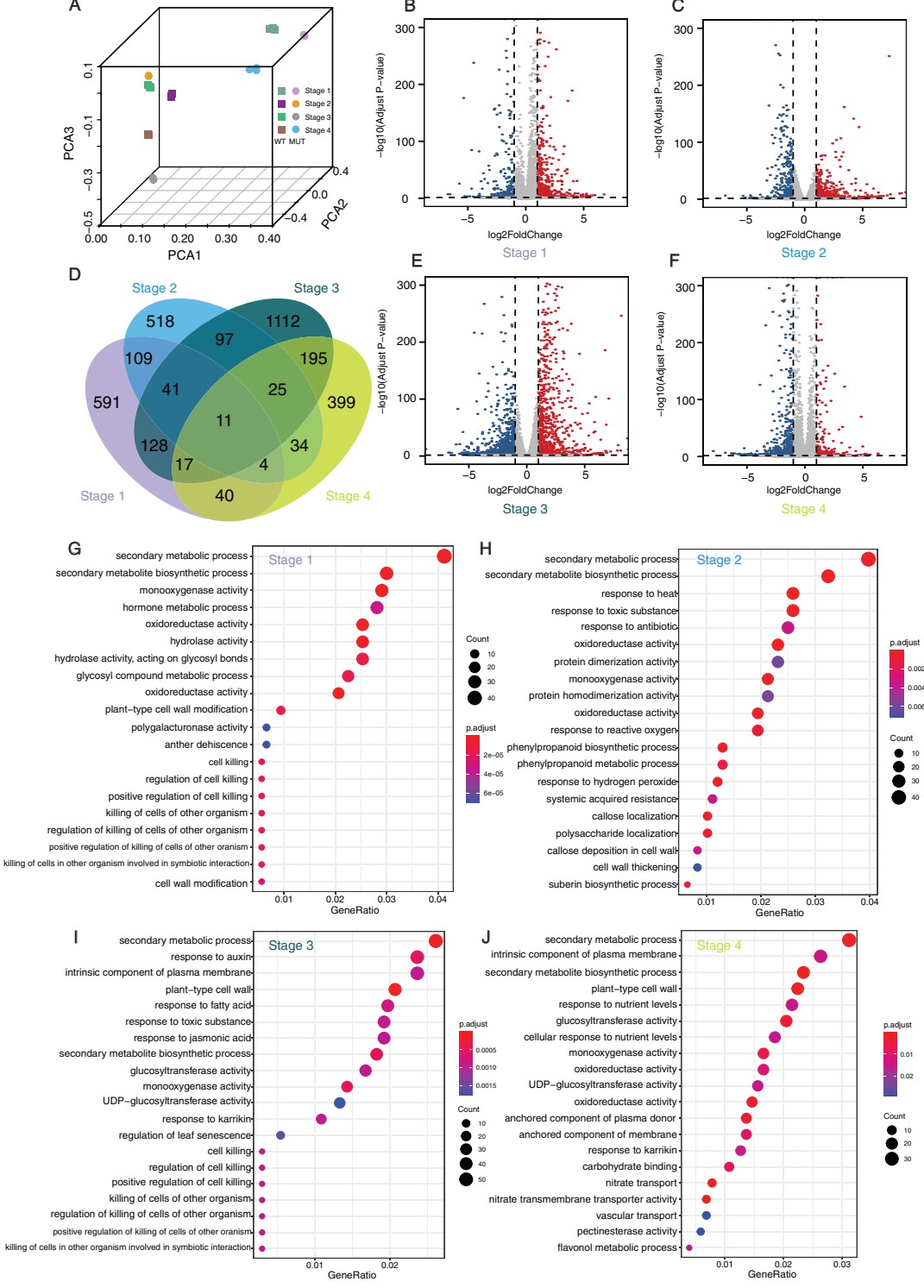

**Figure 4 Analyses and statistics of DEGs and GO enrichment.** Analyses and statistics of DEGs and GO enrichment. (A) Principal component analysis (PCA) of transcriptome data. (B, C, E, and F) Volcano plots for expressed genes at different developmental stages. Red dots represented the upregulated genes while blue dots represented the down-regulated genes. (D) Venn diagram of DEGs at four different stages. (G–J) GO enrichment results of DEGs at four development stages. WT: 'Himord Seedless'; MUT: 'Jinzao Wuhe'. Stage 1–4: full-flowering stage, fruit enlargement stage (14 days post-anthesis), véraison stage (35 days post-anthesis) and maturity stage (56 days post-anthesis). The 'Himord Seedless' in the stage 1–4 were the control group and 'Jinzao Wuhe' in the stage 1–4 were the experiment group.
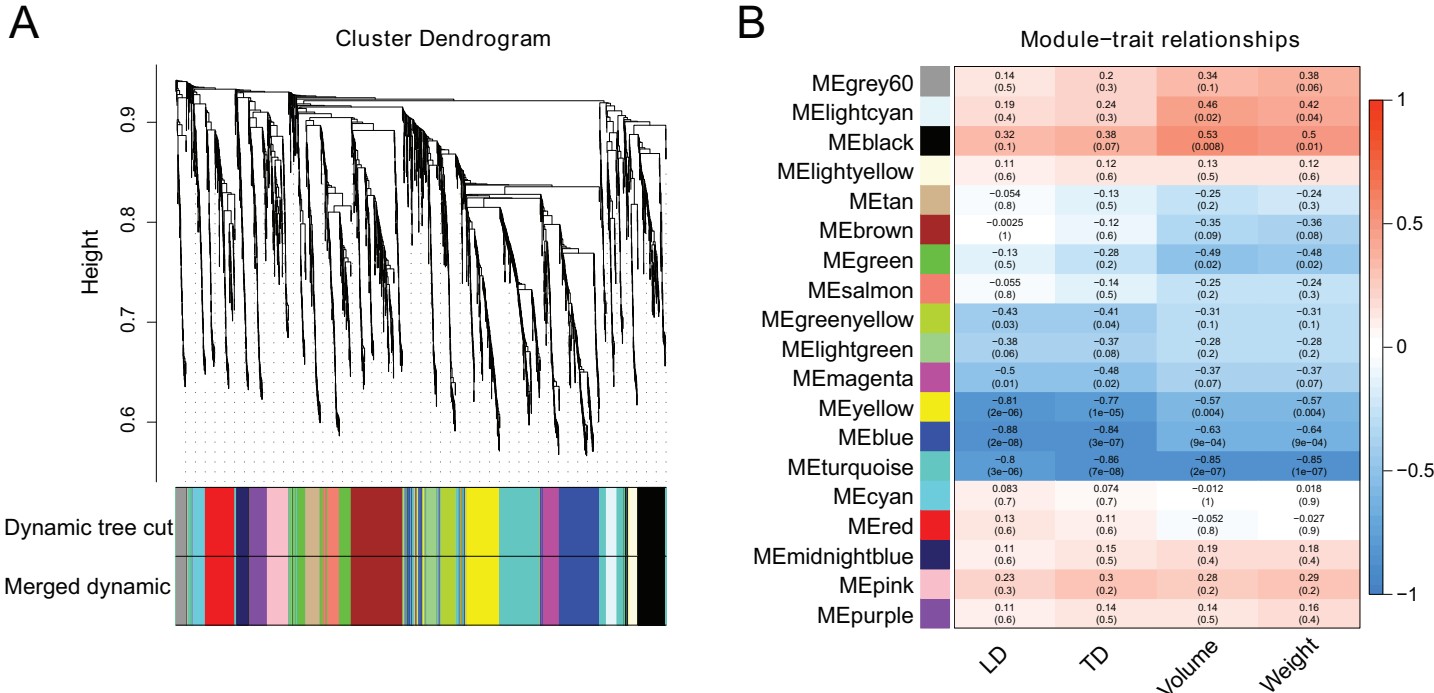

**Figure 5 Weighted gene co-expression network analysis of berry development transcriptomes.** (A) Dendrogram plot of co-expression modules. This plot shows 18 modules and different colors represent different modules. (B) Module-trait relationship and corresponding *p* values. LD: longitudinal diameter; TD: transverse diameter.

"response to fatty acid", "response to toxic substance", "response to jasmonic acid" and "response to karrikin". For DEGs in stage 1 and 3, significantly enriched GO terms were "cell killing", "regulation of cell killing", "positive regulation of cell killing", "killing of cells of other organism" and "regulation of killing of cells of other organism". At the stage 2, the GO terms "phenylpropanoid biosynthetic process", "phenylpropanoid metabolic process" and "suberin biosynthetic process" were significantly enriched. These detected pathways and related genes offered novel clues and contributed to our better understanding of the mechanisms taking place behind the phenomenon of bud sport mutation which leading to berry enlargement in 'Jinzao Wuhe'. (Figs. 4G–4J).

## Identification of genes related to berry enlargement using WGCNA analysis

WGCNA analysis was performed utilizing the FPKM values of 2,984 DEGs to investigate co-expression patterns and determine the hub genes from DEGs. When the soft threshold β was set to 2, the gene network initially corresponded to a scale-free network (Fig. S1). Based on their co-expression patterns, all genes were further grouped into 18 modules (Fig. 5A). Berry phenotype data was used for correlation analysis including transverse diameter, longitudinal diameter, volume and weight. As a result, the 'blue' and 'brown' modules (*p* < 0.01), corresponding to 816 and 333 genes, were significantly clustered and linked to berry enlargement (Fig. 5B; Table S9).

### Integrated results of DEGs and WGCNA for bud sport grapevine 'Jinzao Wuhe'

Based on the previous results from WGRS, RNA-seq and WGCNA, key genes were gradually determined by means of DEGs, specified DNA variations and co-expression pattern with berry phenotypic data. Finally, 42 key genes carrying high or moderate SNPs, InDels and SVs were discovered, which closely related to cell wall modifications (*LOC100258861*, *LOC100258876*), cytochromes P450 (*LOC100244431*, *LOC100253660*, *LOC100266067*, *LOC100254242*), UGT (*LOC100242998*, *LOC100260618*), zinc finger protein (*LOC100260826*), RING-type E3 ubiquitin transferase (*LOC100250551*), ARFs (*LOC100265555*), NAC domain-containing protein (*LOC100251801*), ABC transporter C family member (*LOC100259590*), LRR receptor-like serine threonine-protein kinase (*LOC100243949, LOC100247557, LOC109124293*), protein kinase (*LOC100853763, LOC100258962, LOC100267572, LOC100266874, LOC100258928, LOC104880223, LOC100244602, LOC100246192*), mitochondrial transcription (*LOC100242048, LOC100245527, LOC100853897, LOC100264034*) and other functions (Fig. 6, Table S10).

## DISCUSSION

In grapevine, very few studies have been focused on the berry enlargement through the identifications of genetic variation, quantitative trait loci (QTL) and candidate genes (*Doligez et al., 2002*; *Mejía et al., 2007*). So far, there are very limited materials available for exclusively dissecting the regulatory mechanisms of berry development, especially in bud sport mutation. *Doligez et al. (2002)* detected QTLs related to berry weight using F1 progenies from a cross between two partially seedless genotypes. *Muñoz-Espinoza et al. (2020)* discovered a number of informative and transferable SNP and InDel markers, as well as 68 candidate genes linked with berry size using 'Ruby' and 'Sultanina' cultivars. *Wang et al. (2022)* underlined candidate genes for berry related traits using 'Moldova' (*V. labruscana* × *V. vinifera*) and 'Ruiduxiangyu' cultivars. These varieties are seedless with enlarged berries. In this study, the seedless cultivars 'Jinzao Wuhe' and 'Dawuhebai' are ideal objects with enlarged berries due to bud sport mutation that derived from 'Himord Seedless' and 'Thompson Seedless' cultivars. Especially in 'Jinzao Wuhe', the berries were enlarged nearly twice bigger than its maternal plant, making it an excellent breeding material for studying the regulatory mechanisms underlying berry enlargement.

WGRS is a powerful strategy for genomic variation detection, which could provide direct evidence for phenotypic alterations in plants mainly including SNPs and InDels (*Horton et al., 2012*). Moreover, structural variations (SVs) could be also identified by using Illumina paired end reads and offer clues to the molecular basis of bud sport mutations. In apples, for instance, abundant SNPs and InDels were discovered in various bud mutants and other variants (*Lee et al., 2016*; *Wang et al., 2020*). In grapevines, WGRS of 'Summer Black' and its bud mutant 'SBBM' identified 635 genes carrying differential SNPs and InDels associated with the early-ripening trait (*Xu et al., 2016*). 'Kyoho', 'Zana' and their bud mutants 'Fengzao' and '90-1' were analyzed separately by *Pei et al. (2021)* to identify variations that could explain the altered phenotype between bud sport mutations and their maternal plants in terms of early maturing trait. Herein, WGRS of 'Himord

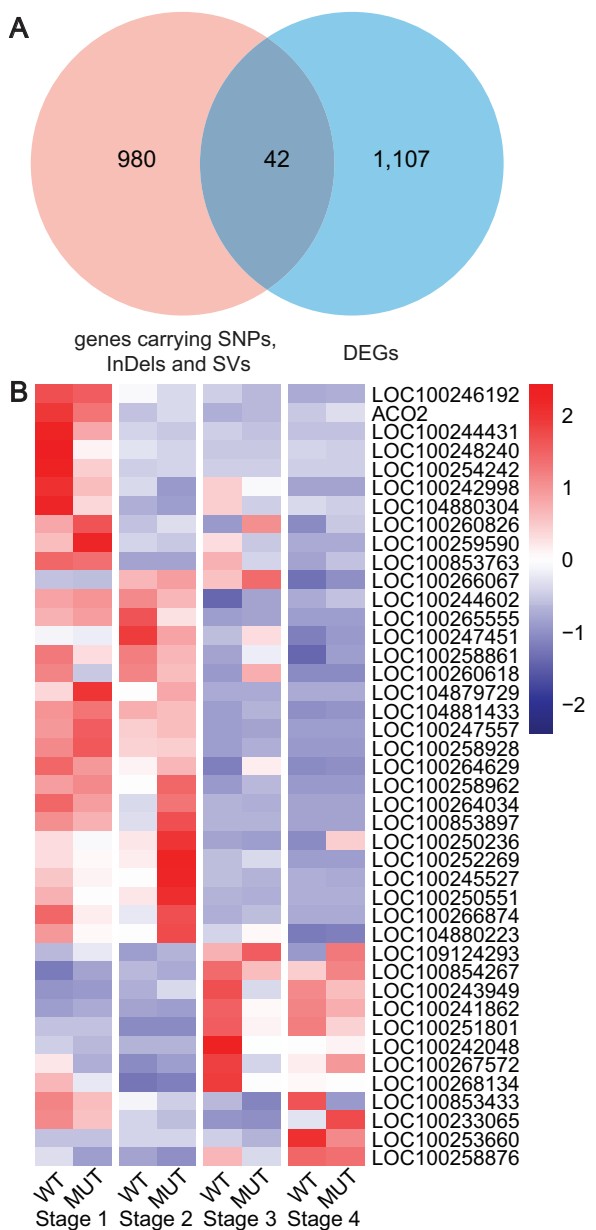

**Figure 6 Analyses and statistics of DEGs carrying specified SNPs, InDels and SVs.** (A) The intersection between DEGs and genes containing SNPs, InDels and SVs. (B) Expression profiles of DEGs harboring specified SNPs, InDels and SV.

Seedless' and 'Thompson Seedless' and their bud mutants 'Jinzao Wuhe' and 'Dawuhebai' resulted in the identification of 1,022 genes carrying differential genetic variants. Combined with the GO enrichment results, in this case, symbiotic microorganisms and plant-type hypersentive response might be the reason or consequence for berry enlargement caused by bud mutation in grapevines (*Marrano et al., 2018*).

Transcriptome sequencing and WGCNA are also practical methods for characterizing and quantifying the gene expression and exploring the potential genes and mechanisms of agronomic traits. RNA-Seq of bud mutants 'Tiangong Moyu', 'Fengzao' and their maternal

'Summer Black', 'Kyoho', respectively, at different stages of berry development led to uncovering potential causal genes regulating ripening period (*Wei et al., 2020*). With the aid of examining gene expression profiles, *Adetunji et al. (2019)* discovered regulatory genes that either delay or promote berry development at the véraison stage of 'Pinot Noir' and 'Pinot Noir Precoce' grapes. *Muñoz-Espinoza et al. (2020)* combined genetic variation detection and transcriptomic approaches to find functional genes involved in berry size and prematurity and develop molecular markers based on SNPs, InDels and SVs, demonstrating the practicability and applicability of RNA-seq. Herein, we discovered 1,149 DEGs associated with berry size based on WGCNA and GO functional categories between 'Jinzao Wuhe' and 'Himord Seedless'. These results indicated that cell wall modification, secondary metabolic processes, stress responses and cell death regulation might play significant roles during berry enlargement (*Sweetman et al., 2012*).

The combination of high-through sequencing and QTL mapping is a practical strategy for screening candidate genes for important agronomic traits in plants. In a very recent study, 772 candidate genes associated with berry size in table grapes have been identified by QTL approach (*Wang et al., 2022*). We compared the results with 1,022 genes harboring SNPs, InDels and SVs and 2,984 DEGs identified in this study by WGRS and RNA-seq approaches. As a result, 81 genes were found in the reported QTL regions including 'RING-type E3 ubiquitin transferase' (*LOC100266002, LOC100250551*), 'Probable xyloglucan endotransglucosylase/hydrolase protein 23' (*LOC100266747, LOC100251303, LOC100246166, LOC100263324, LOC100241056*), 'Expansin' (*LOC100261426, LOC100253046*), 'Auxin-induced protein PCNT115' (*LOC100252909*), 'Auxin efflux carrier component' (*LOC100256460*), 'beta-1,3-galactosyltransferase' (*LOC100246152*), 'Ethylene-responsive transcription factor' (*LOC100259767*), 'LRR receptor-like serine/threonine-protein kinase' (*LOC100241613, LOC100244275, LOC100244999, LOC100262226*), 'Heat shock protein' (*LOC100264870, LOC100252916, LOC104879973*), 'Cytochrome P450' (*LOC100251316*), 'MATE efflux family protein' (*LOC100242781, LOC100263305*) and several genes involved in 'Cell wall modifications' (*LOC100267301, LOC100252982, LOC100242036, LOC100252799*) (Table S11). These genes offered imported clues for understanding the mechanisms of berry development which had been reported to be functionally related to berry size in grapes or other plants (*Xia et al., 2013*; *Doligez et al., 2013*; *de Jong et al., 2015*; *Muñoz-Espinoza et al., 2016*, *2020*).

Pectin and cellulose are key substances involved in the mechanical strength of the primary cell wall and are important to the physical properties of plant cell. The degradation of pectin during ripening seems to be responsible for tissue softening, as reported in several fruit plants including kiwifruit (*Actinidia deliciosa*), lemon (*Citrus limon*), apple (*Malus pumila Mill*) and grapevines (*Vitis vinifera* L.). Moreover, previous studies demonstrated that the cellulose-hemicellulose networks and related genes could regulate cell wall modifications and cellulase degradation, leading to decreased rigidity in grape berry during ripening (*Fasoli et al., 2016*). In this study, PE and glycosyl hydrolase may catalyze the hydrolysis of pectin, resulting in soft berries during the expansion stage of berry development. In rice, UGT has been demonstrated to regulate grain size by modulating cell proliferation and expansion (*Dong et al., 2020*). The expression of UGT

had been shown to gradually decrease with fruit ripening, regulating the early stages of berry development. In our results, the expression levels of two 'UDP-glycosyltransferase-related genes' (*LOC100266067* and *LOC100254242*) significantly varied between MUT and WT. This is consistent with the reported expression trend, suggesting their potential roles in regulating berry size through cell proliferation or differentiation. In addition, 'arabinogalactan protein (AGP)' is highly complex cell wall glycoprotein that plays a role in cell expansion and division (*Leszczuk et al., 2020*). As a result, AGP (*LOC100252982*) was detected to have higher expression in MUT than in WT, with specific accumulation in the second stage of berry development, implying its regulatory role during berry expansion. However, these genes need to be further investigated and functionally validation.

The CYP superfamily genes were also reported to positively associated with fruit size (*Qi et al., 2017*). Four CYP genes (*LOC100244431, LOC100254242, LOC100258062, LOC100266067*) were identified in this study, of which *LOC100258062* and *LOC100266067* were upregulated at berry expansion stage in MUT, implying their functional roles in enlarging berry size in MUT. Similar phenomenon was also discovered in Arabidopsis that overexpression of *AtCYP78A9* resulting in enlarged fruit organs (*Ito & Meyerowitz, 2000*) In addition, auxin was regulated by NAC, which thereby triggered the expression of ARFs (*Tello et al., 2015*; *Li et al., 2022*). Ethylene coupled with increased ACO transcription could produce a modest peak during berry ripening and promote larger berry diameter. Simultaneously, zinc finger protein(*Zhao et al., 2021*), ABC transporter C family member, RING-type E3 ubiquitin transferase (*Xia et al., 2013*), multi antimicrobial extrusion (MATE) (*Suzuki et al., 2015*) mitochondrial transcription, leucine-rich repeat receptor-like protein kinase (*Wang et al., 2019*), F-box kelch-repeat protein (*Li et al., 2011*) and protein kinase (*Ye et al., 2017*) resulted in the enlargement for berries. This result is in agreement with genes related to fruit enlargement between pears, peaches, and strawberries (*Pei et al., 2020*). However, limited results supported simultaneously by evidence at genomic and transcriptomic levels suggested that bud sport mutation might be more complex than previously assumed.

## CONCLUSIONS

In conclusion, combined with WGRS, RNA-seq approaches and integrated bioinformatic analysis, key genes including pectin esterase, cellulase A, CYP, UGT, zinc finger protein, ARF, NAC TF, protein kinase were identified in 'Himord Seedless' and its bud sport mutant 'Jinzao Wuhe'. These results could partially explain its phenotypic changes, especially for berry enlargement in bud sport 'Jinzao Wuhe'. However, DEGs during berry development and DNA variations between bud sport and its maternal cultivar could not agree well with each other, suggesting the complexity of mechanisms underlying bud mutation. Our findings offer new clues for a better understanding of bud sport in grapevines and provide useful information for table grape improvement through molecular breeding.

## ACKNOWLEDGEMENTS

We would like to thank Dr. Ming Li at Zhengzhou Fruit Research Institute, Chinese Academy of Agricultural Sciences, for helping to provide materials of 'Thompson Seedless' and 'Dawuhebai'.

### Funding

This research was funded by the China Agriculture Research System of MOF and MARA, grant number CARS-29, and the Innovative Research and experimental Projects for young researchers of Tianjin Academy of Agricultural Science, grant number 20211002.
The funders had no role in study design, data collection and analysis, decision to publish, or preparation of the manuscript.

### Grant Disclosures

The following grant information was disclosed by the authors:
China agriculture research system of MOF and MARA: CARS-29.
Tianjin Academy of Agricultural Science: 20211002.

### Competing Interests

The authors declare that they have no competing interests.

### Author Contributions

- Jianquan Huang performed the experiments, analyzed the data, authored or reviewed drafts of the article, and approved the final draft.
- Guan Zhang performed the experiments, analyzed the data, prepared figures and/or tables, authored or reviewed drafts of the article, and approved the final draft.
- Yanhao Li analyzed the data, prepared figures and/or tables, authored or reviewed drafts of the article, and approved the final draft.
- Mingjie Lyu conceived and designed the experiments, analyzed the data, prepared figures and/or tables, authored or reviewed drafts of the article, and approved the final draft.
- He Zhang conceived and designed the experiments, authored or reviewed drafts of the article, and approved the final draft.
- Na Zhang conceived and designed the experiments, authored or reviewed drafts of the article, and approved the final draft.
- Rui Chen conceived and designed the experiments, authored or reviewed drafts of the article, and approved the final draft.

### Data Availability

The whole-genome resequencing data set generated and analyzed in the current study are available at NCBI BioProject: PRJNA795039 and PRJNA872374.

All raw RNA-seq data set generated and analyzed in the current study is available at NCBI BioProjects: PRJNA795246, PRJNA872374.

## Supplemental Information

Supplemental information for this article can be found online at http://dx.doi.org/10.7717/peerj.14617#supplemental-information.

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
