# Peer review of "Integrative genomic and transcriptomic analyses of a bud sport mutant ‘Jinzao Wuhe’ with the phenotype of large berries in grapevines"

_PeerJ, doi:10.7717/peerj.14617_

## Round 0.1 · original submission · Major Revisions

Dear Authors,

Your manuscript has been reviewed by experts in the field and we request that you make major revisions before it is processed further.

Please revise your manuscript according to the reviewers' comments.

Reviewer 1 ·

Basic reporting

In this manuscript, the authors performed a comprehensive analysis of the genetic basis for grape fruit enlargement trait using whole genome sequencing and RNA-seq.
The authors have introduced the known regulators for fruit size and seed size. However, it would be better if the authors can introduce the mechanism in more details, especially for the cell wall related genes.
Overall, the writing is professional. But there are some parts need to be explained or introduced in more details. I have specified some parts that need to be introduced in more details.
The authors mentioned that the “Jinzao Wuhe” is discovered and developed by their lab. It would be better if the authors can briefly describe the cultivation strategy leading to the development of “Jinzao Wuhe”.
The structure of this manuscript conforms to PeerJ standards. The figures are mostly relevant, high quality and well-prepared. But the authors need to specify the error bars in Figure 1.
The authors listed the deposition of the whole genome sequencing (WGS) data and RNA-seq data in the online submission form. However, the authors did not include the deposition information in the manuscript. Please make sure to include the deposition information in the manuscript. Additionally, in table 1, the authors listed 4 sample of WGS, but there are only 2 samples in the deposited WGS dataset. Please include the dataset accession numbers for the other 2 samples, even though these might be from a previous publication.

Experimental design

In the method section of transcriptomic analysis, the authors need specify the cutoff of adjusted P value used for DEGs.

Validity of the findings

In line 335-348, figure 6 and table 2, the authors stated 42 genes are discovered to be candidate genes. But how were these genes selected? From Figure 6A, it seems like these 42 genes were selected as the intersection of genes carrying mutations and DEGs. But what is the biological meaning for this criteria? The authors need to explain in details about the selection criteria and justify why this criteria is chosen.

Reviewer 2 ·

Basic reporting

The manuscript by Huang et al presents a genomic and transcriptomic characterization of vitis vinifera mutants with a characteristic phenotype.
The manuscript is interesting and well written. I am enclosing my comments, to contribute to its improvement.

1) For the study presented here to be FAIR, it is necessary that the authors publish the command lines used to do each analysis (either from BASH, R, and/or any programming language used here), the softwares and the parameters applied. In this way, any researcher who requires it, will be able to reproduce the work. I recommend either attaching the code in a supplementary file, or in a Github repository. Without this it is impossible to publish the work, as it is not reproducible.

2) The raw sequencing data are available from ENA/SRA under accession number PRJNA795039 and PRJNA795246. However, this is not mentioned in the manuscript (only mentioned in the notes to the editor). Authors should indicate the accession numbers in the article. I recommend adding a supplementary table with metadata as well.

3) Since for genomic analysis authors only worked with short illumina reads, the only structural variants (SV) they can call with confidence are deletions (i.e. fragment that is absent in the sample, with respect to the reference), and translocations. To call insertions (i.e. regions that are present in the sample, but not in the reference), you have to do de novo assembly. Then, calling duplications with short reads (and even more so in plants) is unreliable, since short reads fail to resolve repetitive regions, so it is impossible to know whether high coverage is due to duplications or repetitive regions. Something similar happens with inversions. Getting an output from bioinformatics software does not mean that you have to blindly trust it. Remove INS, DEP and INV from SVs, leaving only DEL and BND (translocation). Do a re-analysis of the resulting vcf because many times DELLY calls DEL that are overlapping. In that case, unify.

4) The VCFs containing the SNPs, Indels, SVs generated, filtered and used in this study also constitute a result and should be shared. I recommend either uploading it as supplementary material, or sharing it on a Github. Also, format the headers of those VCFs, so that they are FAIR. I recommend following the guidelines of: https://f1000research.com/articles/11-231

5) I would move Table 2 to supplementary material, given its length.

6) In Figure 4, I fail to understand why in all the vulcano-plots the log10 p-values seem to be bounded at their highest value. Check what is going on, because it is not normal to observe plots bounded in that way in their p-values.

7) In Figure 4, correct the GO legends, because some of them are cut off.

8) According to the logic of the WGCNA (doi/10.2202/1544-6115.1128), the number of samples used here is insufficient to have statistical power to apply this strategy. In a typical high-throughput setting, correlations on fewer than 15 samples will simply be too noisy for the network to be biologically meaningful. Delete reference to WGCNA.

9) Delete Figure 7, as it does not represent a regulatory network.

10) Expand discussion. In this work the authors did multiple studies, however the discussion about them is very limited.

Experimental design

All my comments were listed previously

Validity of the findings

All my comments were listed previously

Additional comments

All my comments were listed previously

---

## Round 0.2 · Major Revisions

Dear Authors,

The second round of the review process was completed and reports advised major modifications.

There are several concerns that should be addressed before considering this manuscript for publication.

Reviewer 1 ·

Basic reporting

In the revised version of the manuscript titled "Integrative genomic and transcriptomic analyses of a bud sport mutant ‘Jinzao Wuhe’ with the phenotype of large berries in grapevines”, the authors made substantial efforts to address the reviewers’ concerns and improved the quality of this manuscript. The rebuttal letter is impressive, with detailed responses to reviewers’ comments and highlighting of the text changes.
However, there are still some concerns to be addressed.
1. For the use of BWA tool, the authors cited Houtgast et al., 2018, which is a paper about the hardware acceleration, not the original publication about the development of BWA. Please cite the original BWA-MEM paper, Li H. (2013) Aligning sequence reads, clone sequences and assembly contigs with BWA-MEM. arXiv:1303.3997v2, as requested by Heng Li, the author of BWA (https://github.com/lh3/bwa).
2. In Figure 4A, “Principal component analysis (PCA) of transcriptome data”, the authors should specify the samples represented by each datapoint. Also, the authors should describe the generation of PCA plot in the methods section.
3. In Figure 4B-J, please specify in the figure legend that which cultivar was the control group and which cultivar was the experiment group when performing DEG analyses.
4. Some of the references were not formatted correctly. For example, line 591-593, line 648-650, line 654-655, line 658-661, line 735-738. These are journal articles but were cited as webpages hosting these articles. Additionally, in line 369, it should be “Muñoz-Espinoza et al.” instead of “Claudia et al”. Please check the records in the citation management software, to correct these and any additional citation errors.
5. Gene names should be italicized. For example, “VvCEB1” and “VvSAUR1” in line 101.
6. Line 167 and Figure 3 legend, “V.vivfera” should be “V. vivfera”. There was a space missing between the genus and species name.
7. Line 102, should “VviANT1” be “VvANT1”?

Experimental design

No further comments.

Validity of the findings

No further comments.

---

## Round 0.3 · Minor Revisions

Dear Authors,
Reviewers have completed their reports and have advised that there minor modifications which should be addressed in the revised version.

Please ensure that all recommendations have been considered.

Best regards,

Othmane MERAH

Reviewer 1 ·

Basic reporting

In the revised version of the manuscript titled “Integrative genomic and transcriptomic analyses of a bud sport mutant ‘Jinzao Wuhe’ with the phenotype of large berries in grapevines”, the authors revised the manuscript well.
There are a few minor copy-editing errors remained to be corrected. For example, an extra “;” at line 409; the ‘’ should be input with an English keyboard or input method at line 441-445. And there are a few missing spaces in the manuscript. I recommend the authors to perform a careful and word-to-word level proofreading to correct any additional minor errors before publication.
Overall, the authors revised the manuscript well. I congratulate the authors for finishing this study, and recommend acceptance after minor copy-editing.

Experimental design

No further comments.

Validity of the findings

No further comments.

---

## Round 0.4 · accepted · Accept

All requested modifications have been considered and answers to the recommendations have been provided.